# Dynamic Composite Materials Characterisation with Hopkinson Bars: Design and Development of New Dynamic Compression Systems

Mostapha Tarfaoui [1,2]

1 ENSTA Bretagne, IRDL, UMR CNRS 6027, F-29200 Brest, France; mostapha.tarfaoui@ensta-bretagne.fr
2 Green Energy Park (IRESEN/UM6P), Benguerir 43150, Morocco

**Abstract:** The split Hopkinson pressure bars (SHPB) system is the most commonly employed machine to study the dynamic characteristics of different materials under high strain rates. In this research, a numerical investigation is carried out to study different bar shapes such as square, hexagonal, and triangular cross-sections and to compare them with the standard cylindrical bars. The 3D finite element model developed for circular cross-sectional shapes was first validated with the experimental results and then compared with the other proposed shapes. In most scientific research, cylindrical cross-section bars with a square cross-section specimen are traditionally used as they have several advantages, such as in situ imaging of the side surfaces of the specimen during stress wave propagation. Moreover, the flat surfaces of the proposed shapes counter the problem of debonding strain gauges, especially at high impact pressures. Comparison of the results showed an excellent confirmation of the sample dynamic behaviour and different geometric shapes of the bar geometries, which validates the choice of the appropriate system.

**Keywords:** composite materials; split Hopkinson pressure bar; high strain rate; bar shape; material characterisation; dynamic behaviour

## 1. Introduction

J. Hopkinson first performed the experimental procedure with Hopkinson bars, and then his son B. Hopkinson extended the work by developing a theoretical formulation for the full extension of an impulsively charged wire set at one end [1]. About 34 years after Hopkinson developed his pressure bar technique, Davies [2] proposed the first dynamic axial and radial strain measurements in Hopkinson pressure bar experiments employing parallel plates and cylindrical condensers with a double-beam cathode-ray oscillograph. Then, Kolsky [3] published his famous paper on measuring the mechanical properties of several different materials at high loading rates using a modified Hopkinson pressure bar, later known as the Kolsky bar or split Hopkinson pressure bar. This system has been diversely used to examine the dynamic response of various materials under shear, tension, and compression high strain rates [4–6].

It is quite a familiar concept that the sample should deform at a constant rate or be in uniaxial stress equilibrium during the SHPB test because the different dynamic parameters are measured using equations derived from one-dimensional stress wave theory [7,8]. In order to use a one-dimensional approximation of axial wave propagation with good extraction of findings from input and output bars, isotropic materials were evaluated in traditional SHPB tests in a cylindrical shape with an L/D ratio roughly between 0.5 and 2 using different materials [9,10]. Materials, including soft polymers and biological tissues, have been tested at high strain rates using SHPB and their output signal contains less high-frequency noise in the signal because of their self-damping behaviour [11–14].

Conversely, different interpretation and testing protocols were required to determine hard ceramic materials' accurate high-strain rate response [15,16]. Furthermore, studying

the dynamic behaviour of composite materials in high strain rate applications required understanding the mechanical behaviour of the fibres, polymer matrix, and their interfacial bonding [17]. In order to illustrate rate effects, Bo et al. [18] looked at the compressive quasi-static and dynamic response and the damage process of epoxy syntactic foam. Their findings demonstrated that this polymer displayed brittle behaviour in quasi-static and dynamic studies that considered the impact of strain rates ranging from 330 to 550 s$^{-1}$. The scientists explained this behaviour by the softening phenomena brought on by the polymer's damage modes. In quasi-static and dynamic testing, Chen et al. [19] investigated two thermoset polymers' tension and compression behaviours (Epoxy and PMMA). Their findings demonstrated a distinct distinction in the dynamic behaviour and the mode of failure (brittle or ductile) of both polymers under tensile and compression tests. It was also discovered that both polymers' dynamic compressive strengths under dynamic tests were more significant than those under quasi-static loading. Li et al. [20] used the SHPB to investigate the thermomechanical behaviour of two polymers during dynamic compression trials. Their findings demonstrated that the strain rates significantly impacted their dynamic behaviour. In these dynamic studies, there was no temperature change in the early deformation zone, but the thermal softening effects resulted in a significant temperature increase at high deformation levels. Due to their viscoplastic nature and damage from impacts, some of the few accessible references in the literature demonstrate the occurrence of internal heat dissipation in polymers, which could lead to thermal softening and further impair the reactivity of the composite material. El-Habak et al. [21] investigated the mechanical behaviour of three types of woven glass-fibre-reinforced composites under $10^2$ to $10^3$ s$^{-1}$ strain rates (polyester, vinylester and epoxy). They concluded that the vinyester matrix enables the greatest strength. Using SHPB and altering the loading conditions and fibre orientation, Tarfaoui et al. [22] investigated the impact of strain rate on the dynamic behaviour of glass-fibre-reinforced polymer under in-plane and out-of-plane dynamic compression testing. The findings show that the material's dynamic strength for in-plane testing is greatly influenced by fibre orientation and impact pressure and that for out-of-plane tests, the material is significantly sensitive to fibre orientation at the same impact pressure. The dynamic compressive behaviour of [0/90]26 glass/epoxy laminates in-plane and out-of-plane directions was investigated by Arbaoui et al. [23,24]. They concluded that the material is more resistant under out-of-plane stress than under in-plane loading and that the dynamic properties are strain rate sensitive.

However, SHPB has frequently been used to study the dynamic characteristics at high strain rates ($10^2$–$10^4$ s$^{-1}$) for different materials, including ductile and brittle materials. However, the traditional circular cross-section of bars has resulted in a dispersion effect [25–27]. Moreover, the high-frequency oscillation phenomenon, especially in the case of brittle materials, could compromise the accuracy of the experimental results because of their minimal failure strain and the short time of the rise of stress wave would not let it reach the stress equilibrium before the final failure [28–30].

In recent years, a few researchers have been working on eliminating the errors in the results of SHPB and have proposed some modifications to improve the experimental procedure. Woldesenbet et al. [31] studied the effect of different geometries of specimens in SHPB. Samples with square cross-sections presented various advantages during SHPB tests, such as extracting the specimen's strain deformation directly through real-time imaging of the flat surface. Compared with circular cross-sectional specimens, the imaging of square specimens could be processed later for extracting full-field strain distributions, which was possible by examining the wave propagation in square cross-sectional specimens with optimum L/D ratio. David and Hunter [32] presented some guidelines for the selection of the length and geometry of the specimen for one-dimensional wave propagation analysis.

Moreover, some researchers have studied these SHPB configurations using finite element modelling to correlate the theory with the data [33]. For example, Frew et al. [34,35] presented a design proposition of a pulse shaper in the SHPB technique to study hard and brittle materials such as rocks with good accuracy of results. Similarly, Lu and Li [36] vali-

dated the use of pulse shapers in smoothing the waveform of the SHPB system and making the specimen reach its stress uniformity using finite element modelling. Lee et al. [37] investigated the influence of pulse-shaping material's thickness on the incident wave's high time. Abotula and Chalivendra [38] investigated the effect of diameter-to-thickness ratios (t/d) and different kinds of copper pulse shapers on the incident wave to conclude the conditions to achieve the stress equilibrium of samples. Heard et al. [39] also studied the effect of the annular copper pulse shaper on the large diameter bars of the SHPB experiment. Li et al. [40,41] modified the shape of the striker bar to obtain a spindle shape incident pulse to study the mechanical performance of brittle rocks. Cloete et al. [42] suggested a taper striker bar to reach a constant strain-rate loading, while Vecchio and Jiang [43] modified the length of the striker to change the waveform. This modification in the length of the striker bar also improved the dynamic parameters, such as the stress and strain rate results. Moreover, Baranowski et al. [44] studied the influence of the height of the striker bar on the incident wave using finite element modelling with LS-DYNA software.

To our knowledge, to date, no research has been carried out to investigate the influence of changing the form of the bars of SHPB on the dynamic behaviour of the materials. On the other hand, numerous experimental studies have been carried out to study the dynamic performance of different materials using conventional circular cross-sectional bars. However, the cylindrical bars have the disadvantage of being difficult to mount a strain gauge on their surface to record the incident and transmitted wave signals and often present a problem of debonding at higher impact pressures. So, in this context, a study has been carried out by developing a finite element model to evaluate the effect of different shapes of the bars on the behaviour of materials and the accuracy of the results. The finite element model of conventional circular cross-sectional bars was first validated with the experimental results and then compared with the other bar shapes, i.e., triangular, square, and hexagonal. The flat surface not only solves the problem of attaching strain gauges on the surface of bars to ensure more accuracy in results but also results in better real-time imaging of material deformation.

## 2. Hopkinson Bar Theory

The dynamic compression loading of this material was achieved using the SHPB apparatus; see Figure 1. The principle of functioning of such a set-up is well documented in [7,10,17,25]. The SHPB apparatus consists of a striker, an incident bar, and a transmitter bar. This apparatus aims to apply a given impact pressure to the specimens between the incident and transmitted bars and determine its dynamic proprieties at a characteristic strain rate. The SHPB signals are recorded and treated to determine the dynamic properties of the tested material under the dynamic compression experiments at different strain rate levels; see Figure 2.

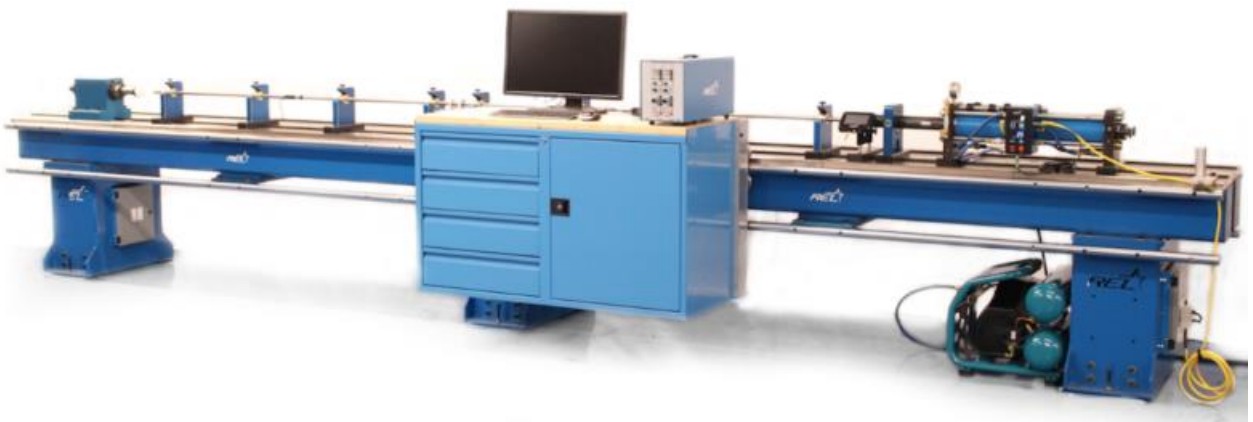

**Figure 1.** SHPB apparatus.

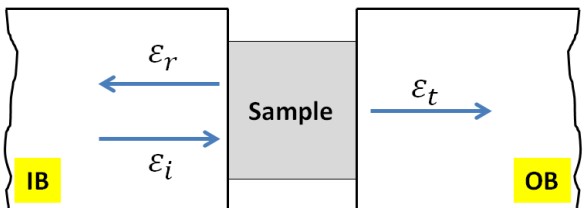

**Figure 2.** Hopkinson bar theory.

The nominal strain rate $\dot{\varepsilon}(t)$ can be determined by considering a homogeneous deformation of the specimen respecting one-dimensional wave theory as:

$$\dot{\varepsilon}(t) = -\frac{2c_0}{L}\varepsilon_r(t) \tag{1}$$

where L is the original gauge length of the specimen, $\varepsilon_r$ (t) is the time-resolved strain associated with the reflected wave in the incident bar (IB), and $c_0$ is the elastic wave velocity of the bar material. The following fundamental relations are hence used to determine the dynamic properties (strain $\varepsilon$ (t), stress $\sigma$ (t), loads F (t), and velocities V (t)) of the material at the given strain rates:

$$\varepsilon(t) = -\frac{2c_0}{L}\int_0^t \varepsilon_r(t)dt \tag{2}$$

$$\sigma(t) = \frac{A_t}{A_s}E_t\varepsilon_t(t) \tag{3}$$

where $A_s$ is the cross-sectional area of the specimen, and $\varepsilon_t$ (t) is the time-resolved axial strain in the output bar (OB) of cross-sectional area $A_t$ and Young's modulus $E_t$.

$$F_i(t) = A_b E_b[\varepsilon_i(t) + \varepsilon_r(t)] \tag{4}$$

$$F_t(t) = A_b E_b \varepsilon_t(t) \tag{5}$$

$$V_i(t) = -C_0[\varepsilon_i(t) - \varepsilon_r(t)] \tag{6}$$

$$V_t(t) = C_0 \varepsilon_t(t) \tag{7}$$

where $A_b$ is the cross-sectional area of the bars, and $E_b$ is Young's modulus of the bar. $\varepsilon_i$ (t) is the time-resolved axial strain in the incident bar.

## 3. Experimental and Numerical Validation

To illustrate the developed model's robustness, the example of the dynamic compression of a glass/epoxy composite is discussed in this section. The material used in this study consists of 2400 Tex E-Glass fibres impregnated with an epoxy matrix. The resin is an EPOLAM pre-polymer, EPOLAM 2020 hardener, and 2020 accelerator from Axson. The reinforcement consists of a plain weave fabric with 90% warp yarns and 10% weft yarns. Panels were made by infusion process, and three orientations were studied: $\pm 20$, $\pm 60$ and 90 °C. The square panels of $500 \times 500$ mm$^2$ were cut into cubic samples of $13 \times 13 \times 13$ mm$^3$. The geometric characteristics are presented in Table 1.

**Table 1.** Characteristics of the composite samples.

| Panel | Thickness, (mm) | Surface (mm$^2$) | Void Fraction (%) | Stacking Sequence | Fibre Volume Fraction (%) |
|---|---|---|---|---|---|
| A | 12.52 (0.3) | $13 \times 13$ | 2.00 | $[\pm 20]_{20}$ | 54.0 |
| B | 13.00 (0.1) | $13 \times 13$ | 1.78 | $[\pm 60]_{20}$ | 55.0 |
| C | 13.00 (0.1) | $13 \times 13$ | 2.26 | $[90]_{40}$ | 53.5 |

Split Hopkinson pressure bar tests were modelled to study the composite materials' stress wave propagation and dynamic deformation. Commercially available finite element software ABAQUS was used throughout the numerical studies. Considering the arrangement shown in Figure 2, both incident and output (transmitted) bars were modelled with a diameter of 20 mm and a length of 3 and 2 m, respectively. Likewise, the striker was 0.5 m in length and the same diameter. These bars are correctly aligned and are able to slide freely on their support. The composite specimen is not attached to the bar to prevent the perturbations of measurements due to additional interfaces. The experimental set-up consists of (1) a stress-generating system which is comprised of a split Hopkinson bar and the striker, (2) a specimen, (3) a stress measuring system made up of sensors (typically, resistance strain gages), and (4) a data acquisition and analysis system. The signals are treated with Maple Software using fast Fourier transformation to obtain the evolution of the dynamic parameters: stress vs. strain, strain rate vs. time, incident and transmitted load and velocity at the interfaces input bar/sample and output bar/sample vs. time. A typical signal of the incident, reflected, and transmitted bars measured from strain gauges and recorded by the digital oscilloscope at the strain rate of 831 s$^{-1}$ test is shown in Figure 3.

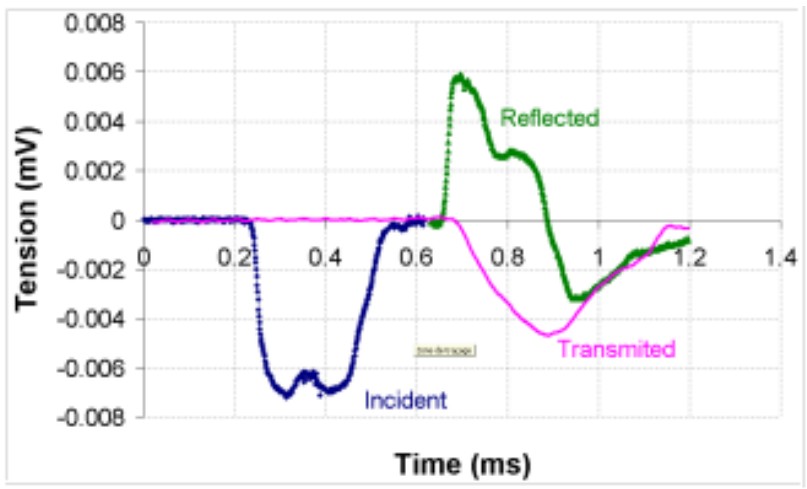

**Figure 3.** Example of incident, reflected, and transmitted pulse, $p$ = 0.9 bar (831 s$^{-1}$).

The incident, transmitted and the striker bars were modelled as an isotropic elastic material. Meanwhile, the specimen was a standard size of 13 × 13 × 13 mm$^3$ and was modelled with an orthotropic elastic material. This composite specimen is made up of 40 stacked plies with a ply thickness of 0.325 mm. An assembly containing all parts (bars, striker, and specimen) was modelled using three-dimensional solid 8-node linear brick elements, with reduced integration and hourglass control (C3D8R in ABAQUS library). The incident and transmitted pulses and the striker bars had uniform mesh with 104,192, 97,870, and 21,890 elements, respectively. The specimen meshed into 8788 elements. The mesh configuration of the composite specimen appears in Figure 4a, while Figure 4b presents detail of the whole model assembly. A surface-to-surface contact is defined at the interfaces of different parts of the SHPB set-up to simulate the interaction at these interfaces, allowing for compressive loads to be transferred between the slave nodes and the master segments. Material properties used in the finite element code are shown in Table 2. The skins with a negligible thickness acting as gauges were placed on the incident and transmitted bars to determine the incident, transmitted and reflected waves. These skins were modelled using the mesh with membrane elements M3D4R (A four-node quadrilateral membrane, reduced integration, hourglass control). Initial velocity conditions were applied to the whole striker volume (all nodes), whose value corresponded to the actual one, e.g., V = 5 m/s. Initial boundary conditions were applied to the striker and the bars such that only movement in one direction was allowed; see Figure 4.

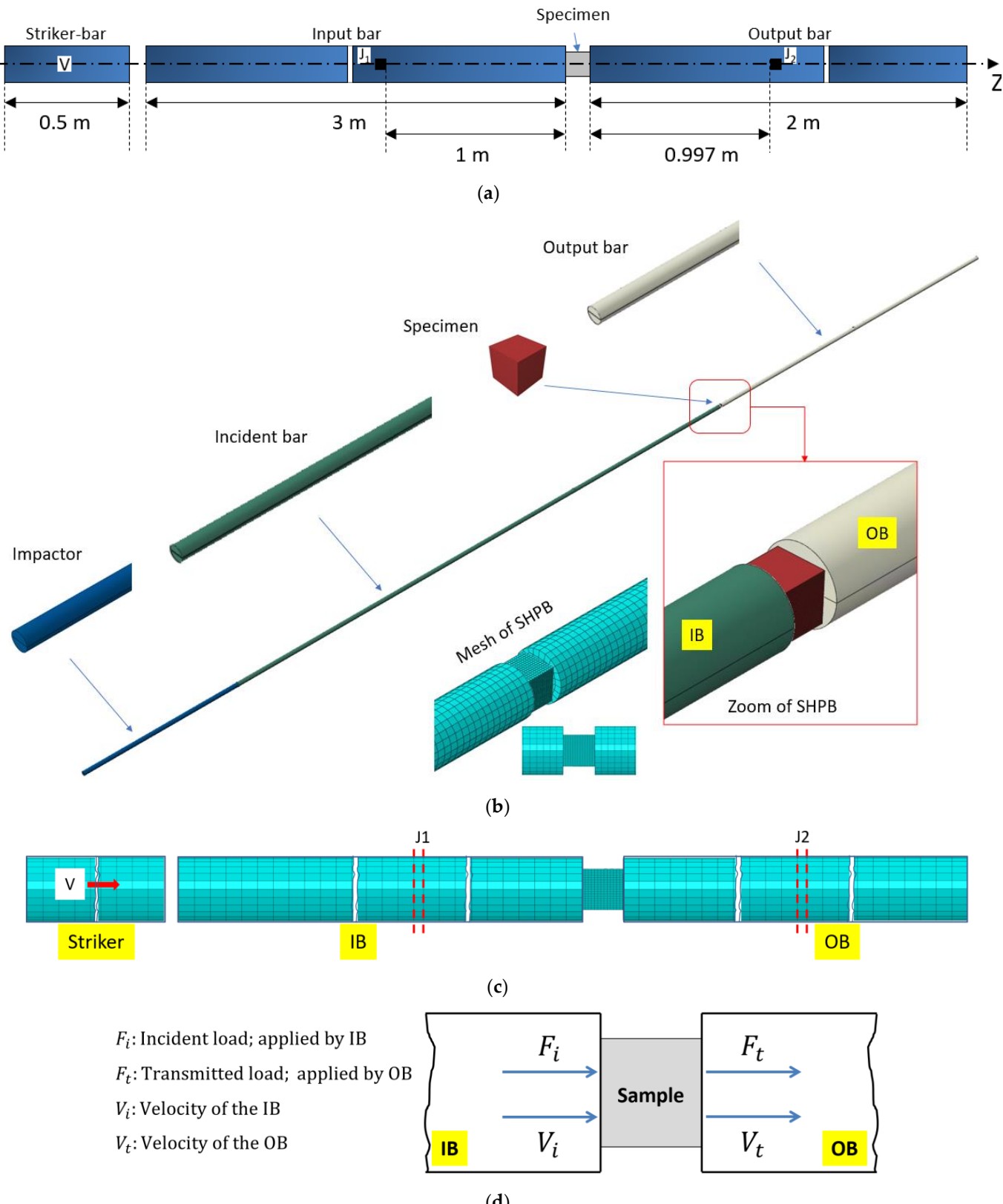

**Figure 4.** FEA of split Hopkinson pressure bars. (**a**) Geometry of the SHPB system. (**b**) Details of the finite element model. (**c**) Mesh and boundary conditions. (**d**) Parameters for the comparison between experimental and numerical results. Test_Inc: experimental incident load, applied by IB ($F_i$). Test_Tran: experimental transmitted load, applied by OB ($F_t$).

**Table 2.** Elastic properties of E-glass/epoxy composite.

| Density $\rho$ (kg/m³) | $E_1$ (MPa) | $E_2$ (MPa) | $E_3$ (MPa) | $\nu_{12}$ | $\nu_{13}$ | $\nu_{23}$ | $G_{12}$ (MPa) | $G_{13}$ (MPa) | $G_{23}$ (MPa) |
|---|---|---|---|---|---|---|---|---|---|
| 1840 | 46,217 | 16,086 | 9062 | 0.28 | 0.41 | 0.097 | 2224 | 3500 | 4540 |

Figure 5 illustrates a comparison between the experimental data and the results of the numerical model. This comparison presents an excellent confrontation for the various orientations. A slight difference between the experimental and numerical results is observed. It should be noted that the absence of the damper bar in the global model causes multiple reflections where the presence of a tensile wave of the same amplitude.

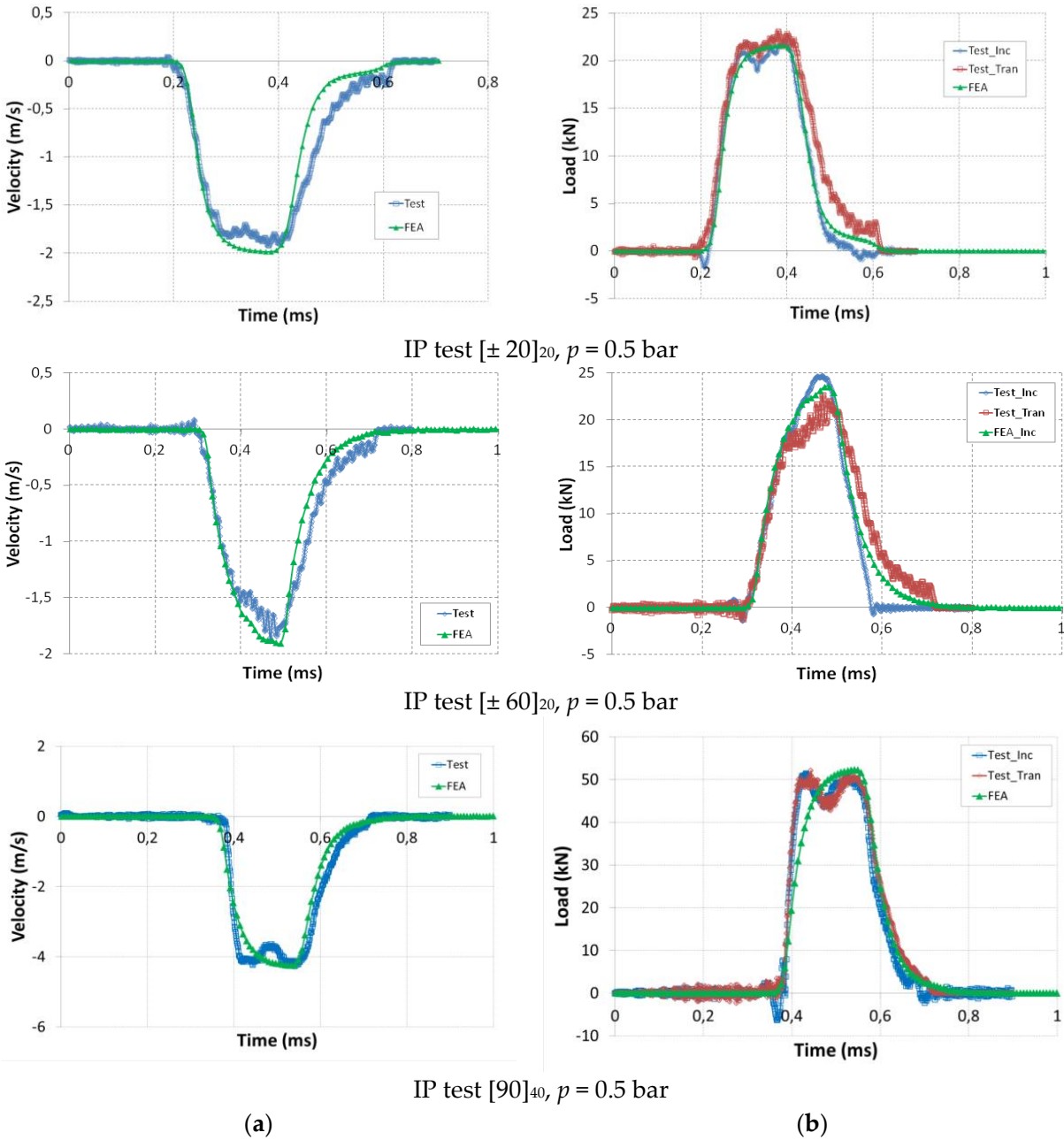

IP test [± 20]$_{20}$, $p$ = 0.5 bar

IP test [± 60]$_{20}$, $p$ = 0.5 bar

IP test [90]$_{40}$, $p$ = 0.5 bar

(**a**)　　　　　　　　　　　　　　　　　　　　　　　　(**b**)

**Figure 5.** Correlation of experimental/numerical results. (**a**) Transmitted velocity. (**b**) Incident and transmitted load.

As part of the dynamic characterisation of an aluminium sample and the quantification of the effect of bar geometry, a comparative study was carried out with four different geometric shapes of the bars. To do this, a non-damaging strain rate was chosen and applied to a cubic-shaped aluminium specimen. Before presenting the numerical results of the dynamic behaviour of aluminium with the different SHPB systems, it was essential to ensure that all simulations were performed under the same conditions, such as the same length of bars, the same boundary conditions, as well as the same mesh so that the results can be comparable. Furthermore, the dimensions of the cross-section of the bars are chosen so that the work is done with the same cross-section A = 314 mm$^2$.

## 4. Parametric Study: Effect of Bar Geometry

The necessity of changing the shapes of an existing protocol for using a circular cross-section of split Hopkinson bars depends on the application. If the application requires a circular cross-section, then it is necessary to modify the protocol. Otherwise, the existing protocol may be sufficient if the application requires a different shape. Circular cross-sections are used for split Hopkinson bars because they provide maximum stiffness and strength while minimizing the amount of material used. The circular shape also allows the bar to be easily split into two halves. The circular cross-section also helps to spread the stress over a larger area, reducing the risk of failure due to a localized point of high stress. Other geometries of a straight section of Hopkinson bars are possible, including curved cross-sections, stepped cross-sections, and hollow cross-sections. However, it is important to note that the design of the bar should be tailored to the specific application as the geometry of the bar affects its performance.

The question that can be asked is: can we use a square cross-section for Hopkinson bars?

The answer to this question is: it is possible to use a square cross-section for Hopkinson bars. The Hopkinson bars are designed to transmit force from one end to the other, so the shape of the bar is not a limiting factor. In fact, square bars can offer the advantage of greater stability and strength when compared to round bars. However, the end fittings used with the bar must be designed to accept the square shape. It is also possible to use a hexagonal or triangular cross-section for the bars. This is because the shape of the cross-section does not affect the functioning of the bar as long as the material remains the same. The split Hopkinson bar works by compressing two bars together and measuring the force that is generated by the strain. The shape of the cross-section does not affect the force produced, and therefore it is possible to use squares, hexagons, and triangles as cross-sections.

The most advanced features currently available in finite element (FE) Abaqus/Explicit have been employed to simulate the behaviour of the composite material under dynamic compression. This article aims to quantify the effect of bar geometry on the material's dynamic response; see Table 3. In the arrangement depicted in Figure 4, the input and output bars and specimen were both modelled with C3D8R elements. The mechanical behaviour was analysed for aluminium and executed in the ABAQUS software. Tables 3 and 4 give the characteristics and properties of the 164 materials used for finite element calculations.

**Table 3.** Elastic properties of the bars and aluminium 6014-T4.

| Material | Young's Modulus E (MPa) | Poisson's Ratio, $\nu$ | Density, $\rho$ (kg/m$^3$) |
|---|---|---|---|
| Steel (striker and bars) | 182,000 | 0.32 | 7800 |
| Aluminium (sample) | 70,000 | 0.35 | 2700 |

**Table 4.** Characteristics of the analysed geometries: input bar, output bar, and striker.

| Shape | Dimension (mm) | Surface (mm²) | Sketch |
|---|---|---|---|
| Circle (SHPB-C) | R = 10 | 314 |  |
| Triangle (SHPB-T) | B = 26.94 H = 23.33 | 314 |  |
| Square (SHPB-S) | c = 17.7246 | 314 |  |
| Hexagon (SHPB-H) | a = 11 H = 22 V = 19.0526 | 314 |  |

### 4.1. Mesh Procedure

The SHPB system with all parts (striker, bars, and specimen) was modelled using 3D solid linear brick elements with eight nodes, reduced integration, and hourglass control (C3D8R). The input, output, and striker bars had a uniform mesh size of 5 mm, while the mesh size of the specimen was 0.5 mm; see Figure 6. A surface-to-surface contact is defined at the different interfaces of the SHPB system to simulate the interaction, thus allowing the transfer of compressive loads between the slave and the master nodes. In this research, the isometric elements are used for meshing with five integration points utilising the Simpson rule. On the other hand, the selected elements have linear interpolation to find better results for impact with the possibility of severe distortions of the elements.

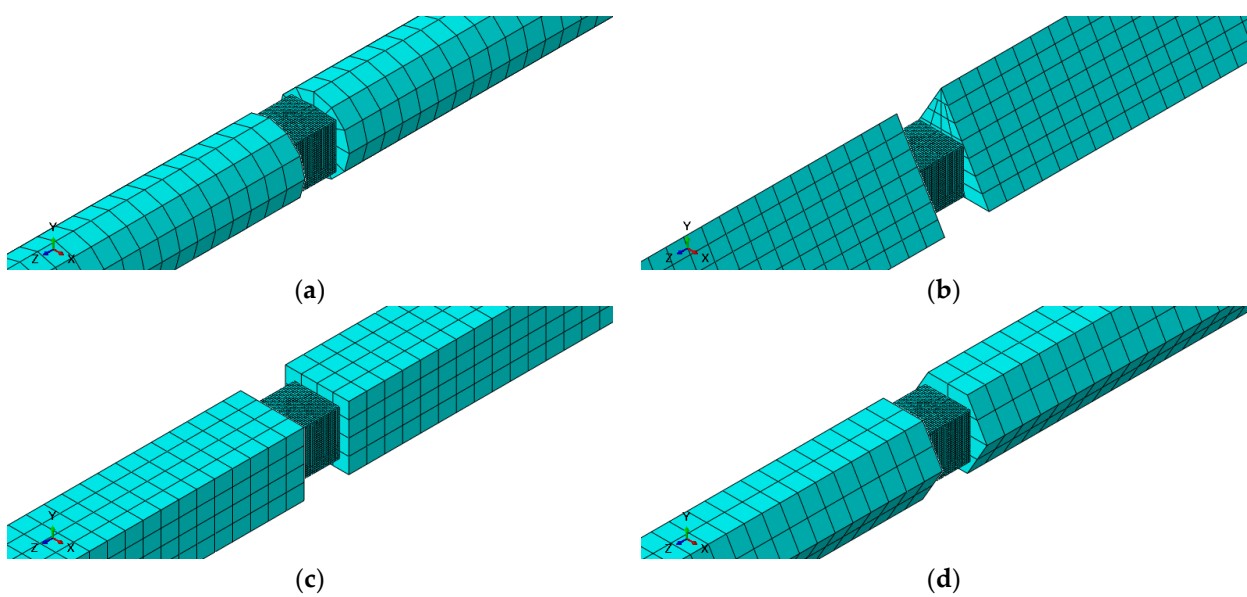

**Figure 6.** Mesh procedure. (**a**) Circle, (**b**) triangle, (**c**) square, (**d**) hexagon.

### 4.2. Results and Discussion

SHPB systems (SHPB-C, -T, -S, -H) with different cross-section geometries were numerically modelled using the finite element software ABAQUS. In addition, the SHPB models with aluminium subjected to dynamic compression were compared. Finally, to quantify the effect of the geometry of the bars and the striker on the response of the material, a technique was set up that makes it possible to compare several parameters:

- Analysis of the incident, transmitted and reflected waves by placing skin elements of negligible thickness used to model the two gauges J1 and J2 placed, respectively, on the input and output bars. These skins were modelled using the mesh with membrane elements M3D4R (four-node quadrilateral membrane, reduced integration, hourglass control).
- The initial velocity conditions were applied to all nodes of the striker volume. The value of this velocity is fixed at V = 5 m/s.
- For initial boundary conditions, only one movement in the z-direction is allowed for the striker and the bars.
- The different physical parameters of strains, velocities, and loads are determined by the numerical model and compared for the different geometries.
- The incident "Fi" and transmitted "Ft" loads are determined, respectively, at the incident bar/sample and transmitted bar/sample interfaces.
- Similarly, the incident "Vi" and transmitted "Vt" velocities are determined, respectively, at the incident bar/sample and transmitted bar/sample interfaces.

Figures 7 and 8 show the state of stresses of the SHPB system with two increments of time, 0.6 ms and 0.7 ms. Overall, the behaviour of the four-bar systems is almost similar. Figure 9 gives the displacement field generated by the different bar systems in the specimen for the same time increments. We have a relatively equivalent distribution for the various SHPB systems; the specimen undergoes almost the same axial displacement. However, we can note that for the hexagonal section, we have a slight difference with the circular section and this difference amplifies with the trianglular section. For a more detailed analysis of the numerical results, we focus on the different parameters of this test of the dynamic compression of aluminium.

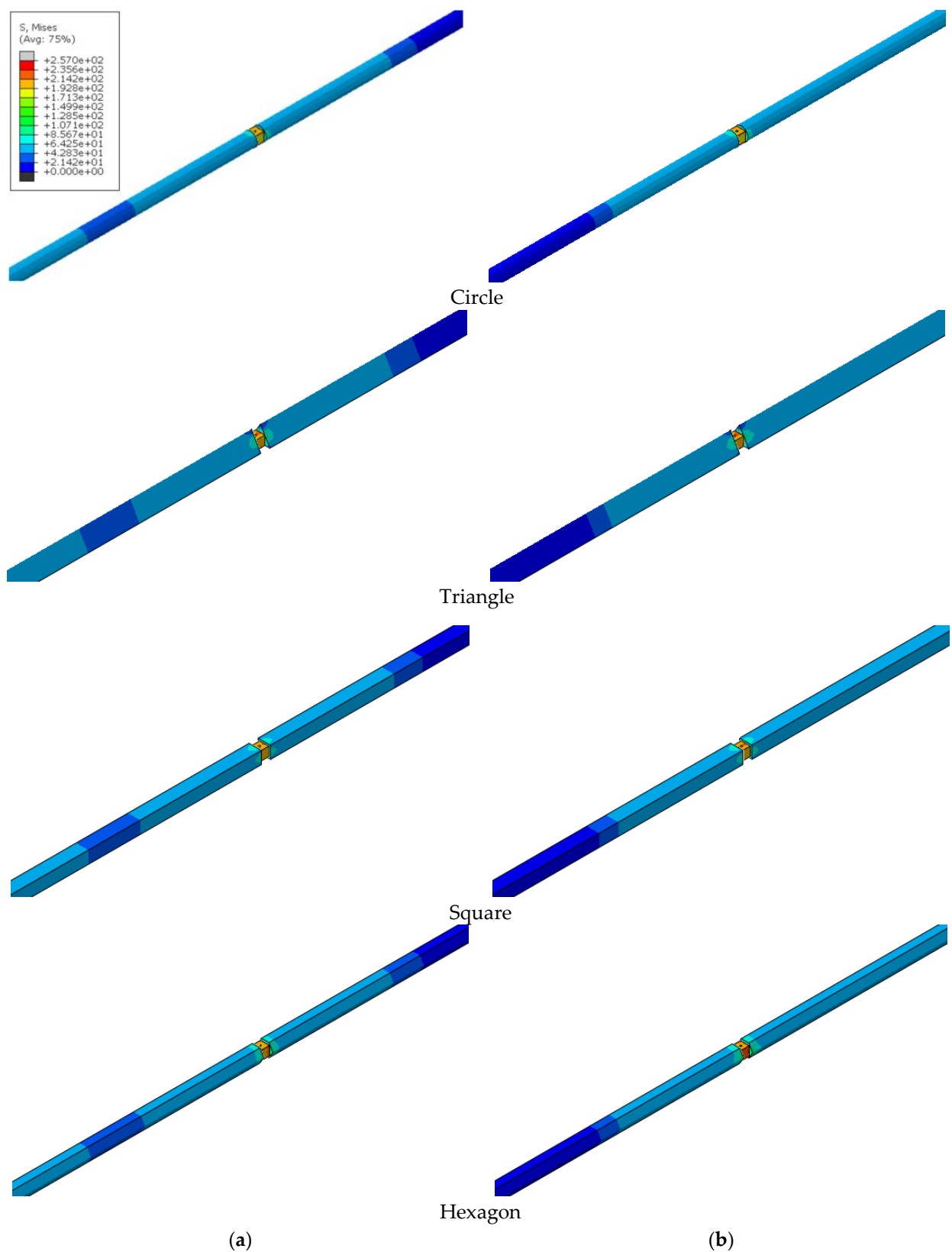

**Figure 7.** State of Von Mises stresses at two increments of time. (**a**) t = 0.6 ms and (**b**) t = 0.7 ms.

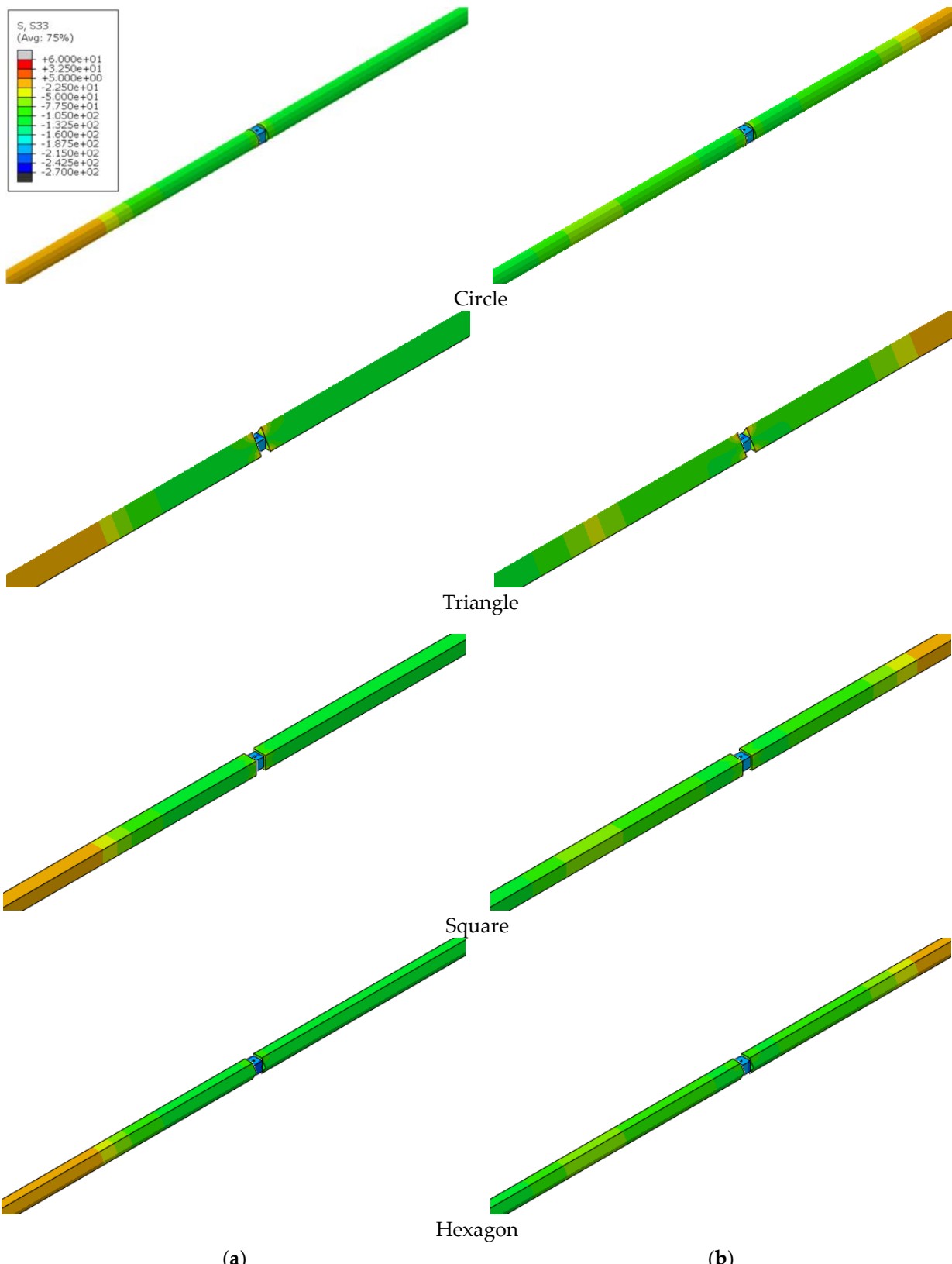

**Figure 8.** State of S33 stresses at two increments of time. (**a**) t = 0.6 ms and (**b**) t = 0.7 ms.

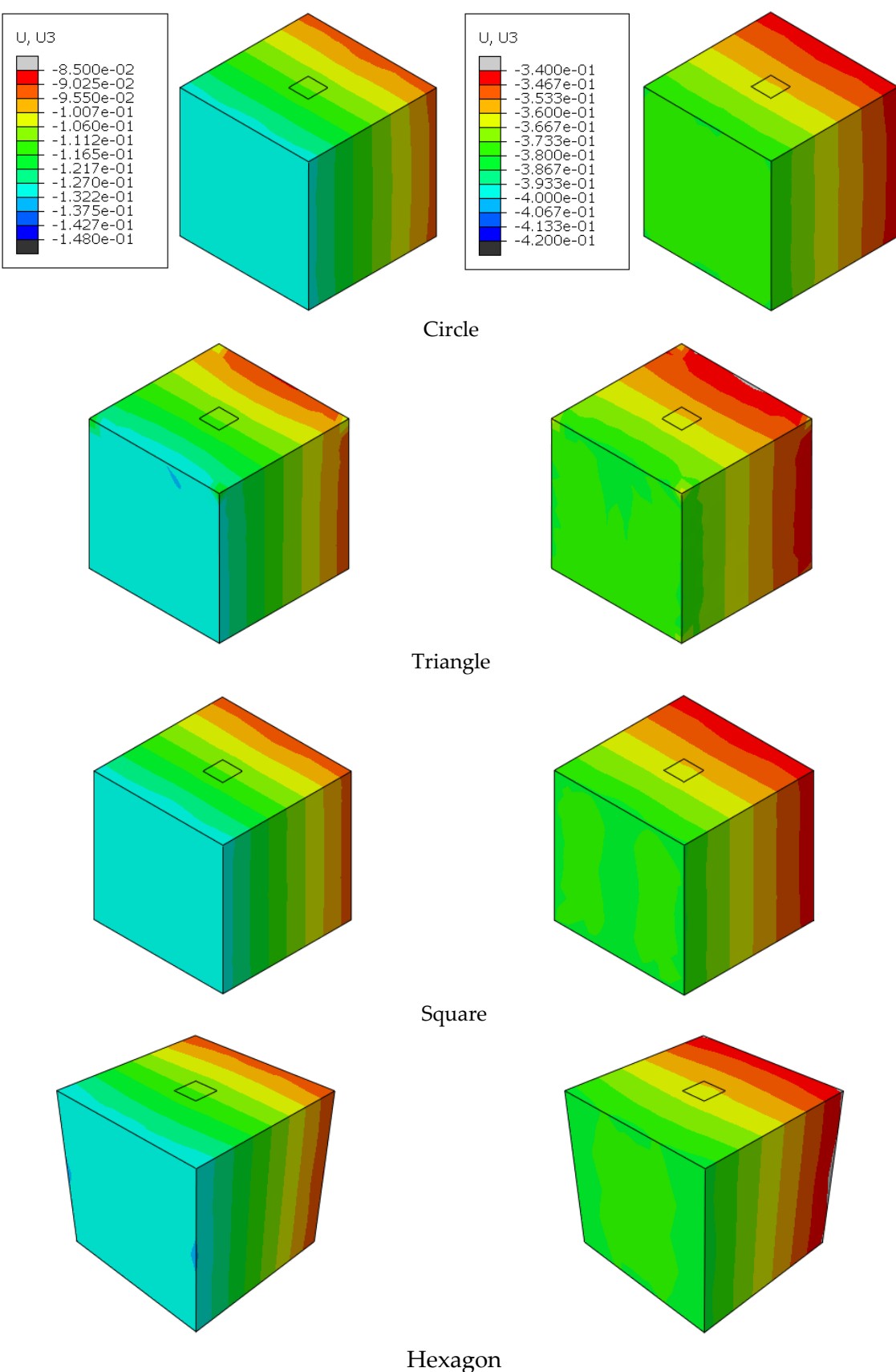

**Figure 9.** Displacement field in the specimen at two increments of time.

The J1 and J2 gauges modelled by M3D4R membrane elements make it possible to find the incident, reflected, and transmitted waves; see Figure 10. It is possible to note the similarity of the signals given by the different systems of bars (circle, triangle, square, and hexagon). For the SHPB-T, the return to zero is sharper than for the other systems; see Figure 10a. In Figure 10b, it is noted that SHPB-T gives a slightly larger transmitted deformation.

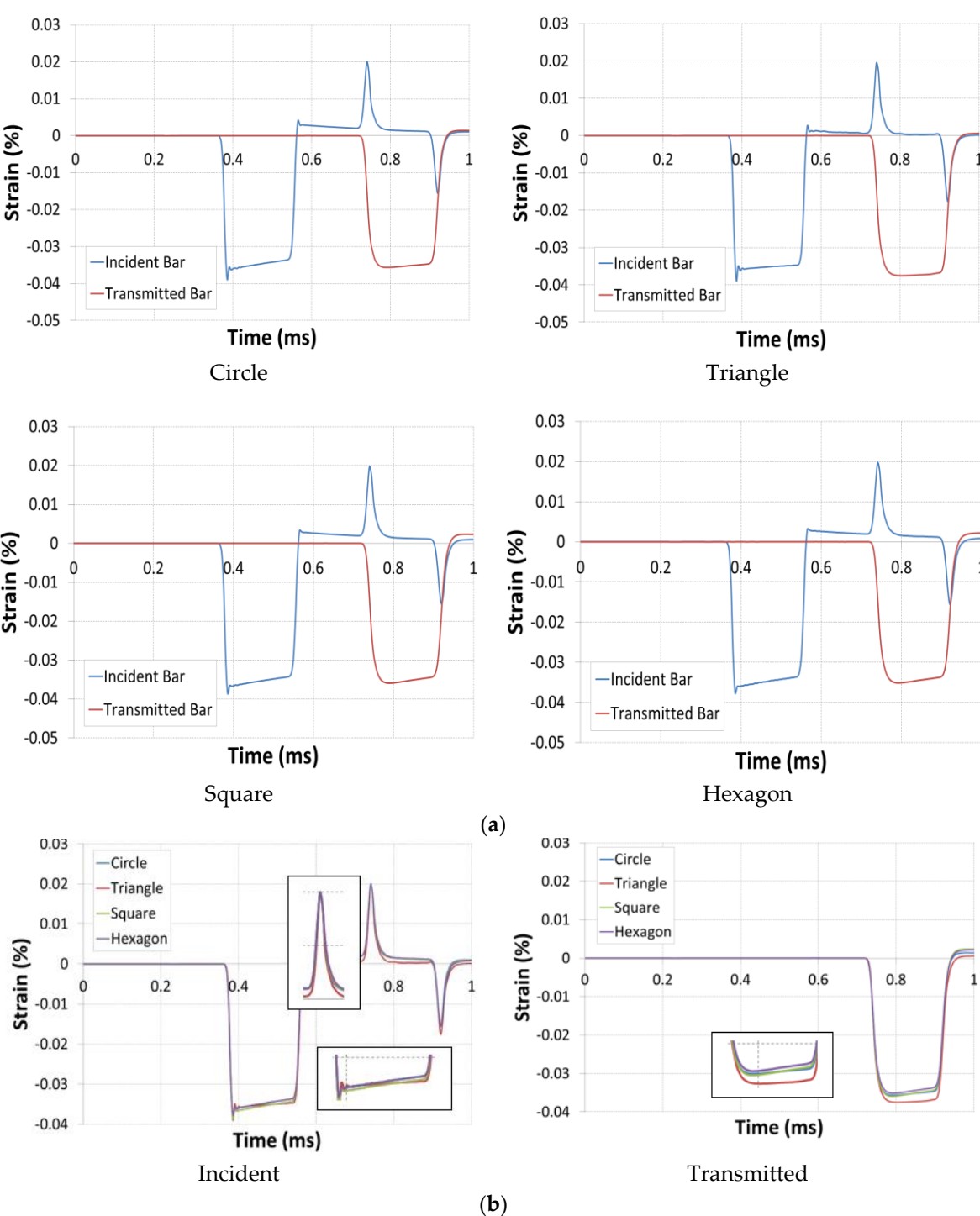

**Figure 10.** Dynamic response of the specimen under the velocity of V = 5 m/s. (**a**) Strain and (**b**) comparative signals.

The curves give the same trend for the incident and transmitted velocity and load at both interfaces, with a slight difference for the maximum values; see Figure 11.

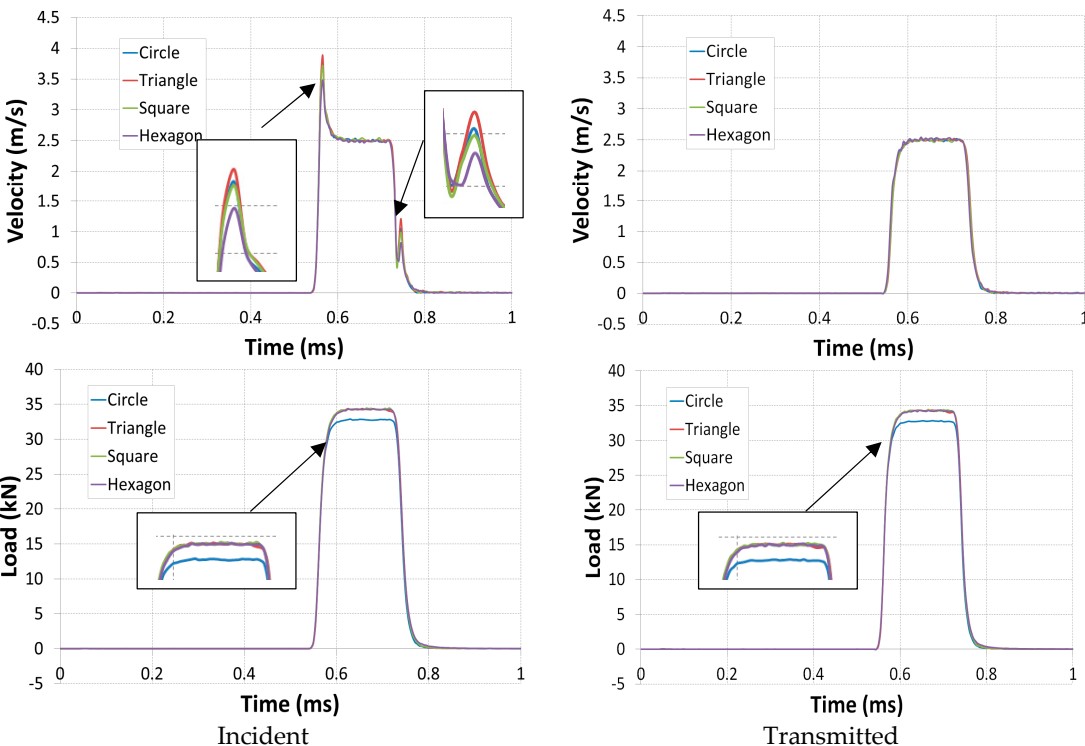

Incident   Transmitted

**Figure 11.** Dynamic compression test parameters with different bar systems.

Now, if one looks at the response of aluminium to this dynamic compression in axial deformation, it appears that the curves have the same trend, but the maximum deformation is slightly different depending on the bar system used. Indeed, the maximum deformations are 0.218, 0.226, 0.237, and 0.252%, respectively, for SHPB-H, SHPB-C, SHPB-S, and SHPB-T systems; see Figure 12.

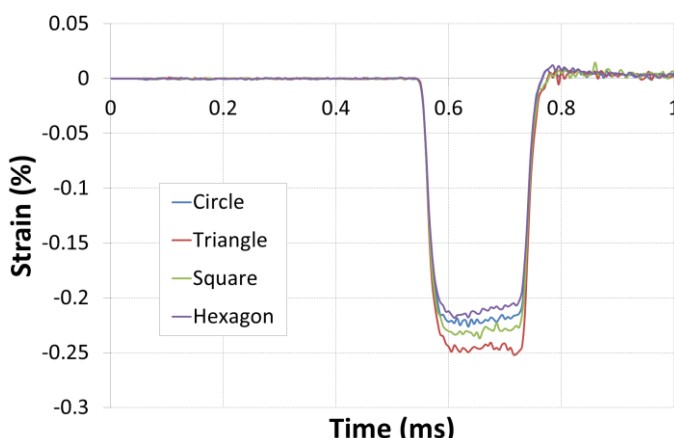

**Figure 12.** Axial deformation of the specimen.

Figures 13 and 14 show the distribution of the pressure exerted on the specimen and the deformation field. We note that the geometry of the bars affects the distribution of these two parameters (pressure and deformation), particularly for the triangular cross-section. Further analysis shows that the material behaves similarly for the circular and square section bars. The behavior laws of Figure 15 show the same trend.

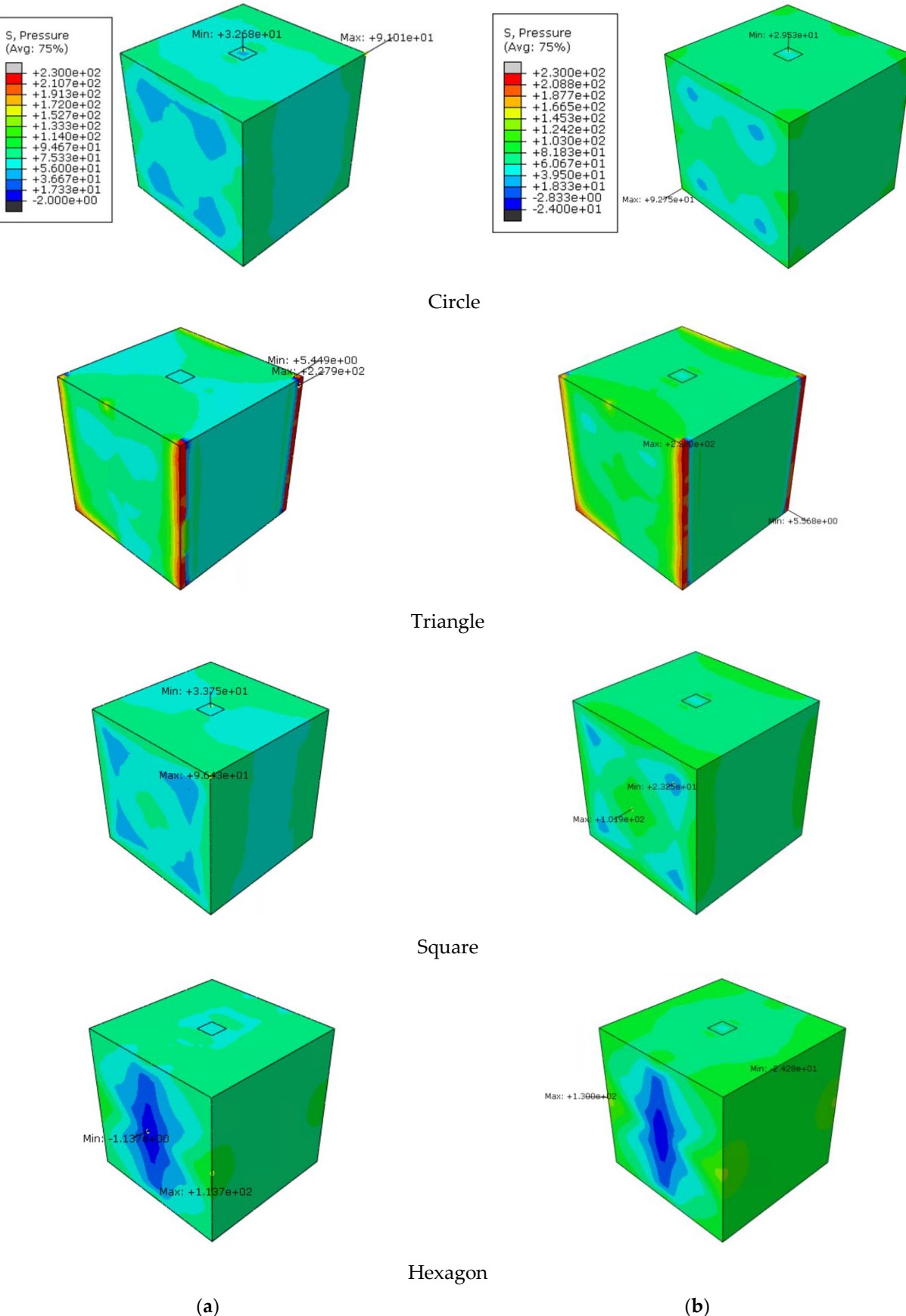

**Figure 13.** Pressure in the specimen at two increments of time. (**a**) t = 0.6 ms and (**b**) t = 0.7 ms.

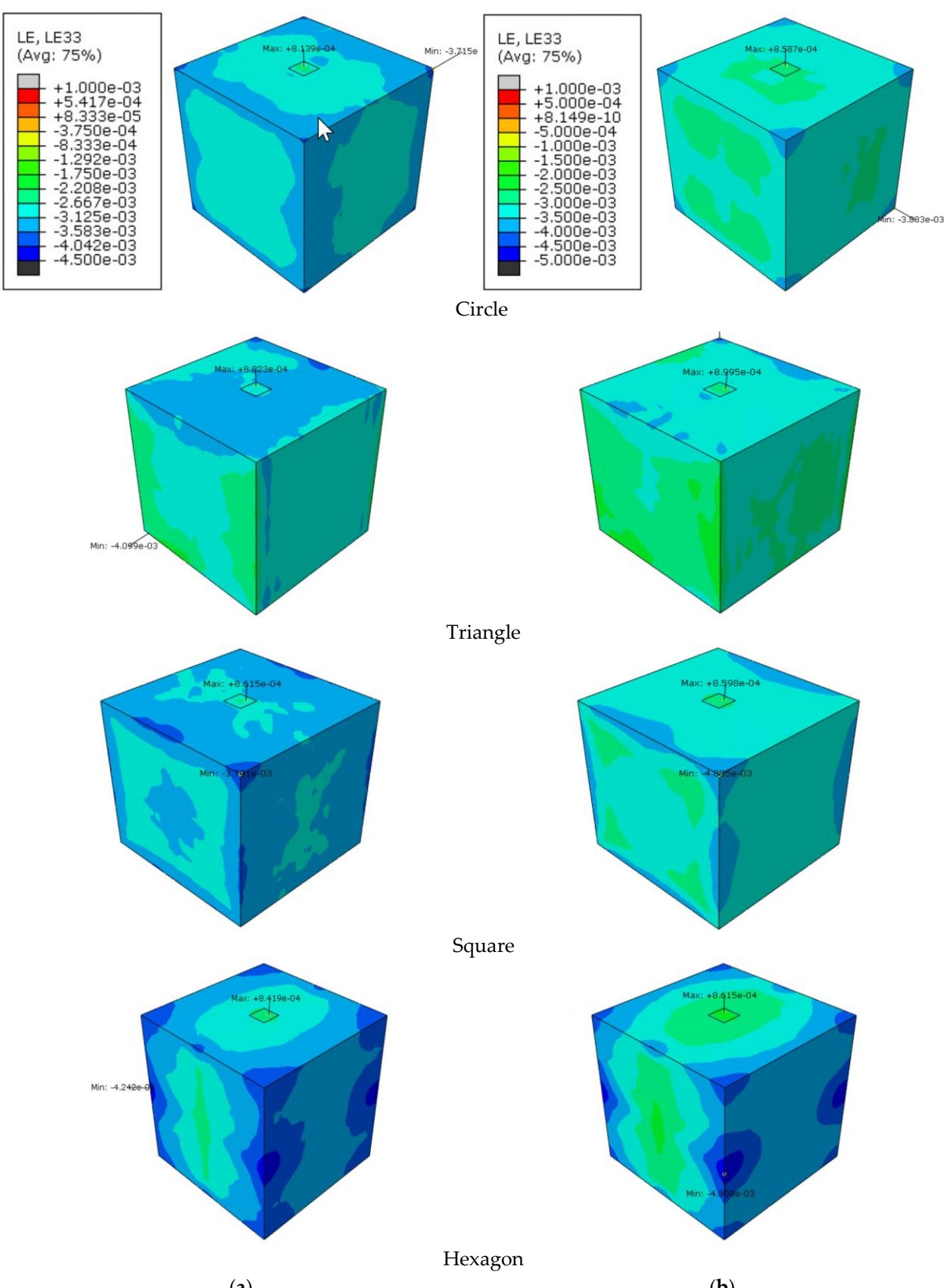

**Figure 14.** Strain field in the specimen at two increments of time. (**a**) t = 0.6 ms and (**b**) t = 0.7 ms.

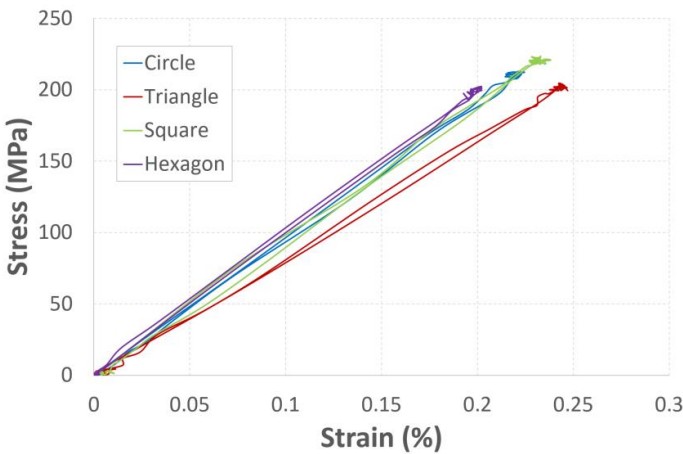

**Figure 15.** Dynamic behaviour under dynamic compression.

Table 5 summarizes the observations concerning the use of the different geometries mentioned above.

**Table 5.** Advantage and disadvantages of the different SHPB systems.

| Shape | Advantage | Disadvantages |
|---|---|---|
| Circle (SHPB-C) | 1. Stronger and more stable than other shapes, making it ideal for high-strain rate testing.<br>2. The circular shape allows for a more uniform distribution of stress throughout the bar, which can result in more accurate strain measurements.<br>3. The circular shape also allows for a more uniform application of force to the specimen, which can lead to more reliable test results. | 1. More expensive to manufacture, making it more cost-prohibitive.<br>2. More difficult to align properly in the testing apparatus.<br>3. Requires more complex fixtures to secure the specimen in place.<br>4. Limits the size of the specimen that can be tested.<br>5. Limits the degree of axial loading that can be applied to the specimen.<br>6. May cause the specimen to fail in a brittle manner.<br>7. May cause the specimen to buckle under axial loading.<br>8. May lead to uneven strain distribution in the specimen.<br>9. May lead to anisotropic behaviour in the specimen.<br>10. May limit the range of strain rates that can be tested.<br>11. Being difficult to mount a strain gauge on their surface.<br>12. Presents problem of debonding at higher impact pressures. |
| Triangle (SHPB-T) | 1. Higher strength-to-weight ratio due to the triangular shape.<br>2. The triangular section can be designed to hold more load due to its higher moment of inertia.<br>3. Easier to form than circular bars. | 1. Can create stress concentrations which could lead to premature failure.<br>2. Results in high friction losses when compared to other cross-sections, such as circular or rectangular.<br>3. High stresses in the bar, caused by the triangular cross-section, can lead to bar failure if the material is not strong enough.<br>4. Can cause poor signal transmission, resulting in a weak signal which can be difficult to interpret.<br>5. More difficult to manufacture than other shapes (high cost). |

**Table 5.** *Cont.*

| Shape | Advantage | Disadvantages |
|---|---|---|
| Square (SHPB-S) | 1. Higher strength-to-weight ratio due to the square shape. 2. The square section can be designed to hold more load due to its higher moment of inertia. 3. Easier to form than circular bars. | 1. The signal-to-noise ratio of Hopkinson bars of the square cross-section is low, making it difficult to measure strain accurately. 2. Stress concentration: The corners of the square cross-section can cause stress concentrations, leading to premature failure of the bar. 3. Difficult to fabricate due to the complex machining required. 4. Limited dimensions due to their complex geometry can limit their use in specific experiments. |
| Hexagon (SHPB-H) | 1. Higher strength-to-weight ratio due to the hexagonal shape. 2. Can be designed to hold more load due to its higher moment of inertia. 3. Can provide a stronger grip on the specimen as it offers more contact points. 4. The specimen can be loaded more evenly as the hexagonal section has a greater surface area than a circular section. 5. Allows for a more efficient distribution of force around the specimen. | 1. Can be more difficult to machine as it requires more complex tooling. 2. Requires more material than a circular section, resulting in increased costs. 3. If the hexagonal section is not machined correctly, it can cause a non-uniform load distribution on the specimen, leading to inaccurate results. 4. It is difficult to maintain a uniform strain rate across the bars due to the irregular geometry of the hexagonal cross-section. 5. The bars are also prone to buckling due to the relatively weaker nature of their hexagonal shape. 6. Hexagonal Hopkinson bars are difficult to fabricate, requiring precise measurements and machining. 7. Additionally, the hexagonal shape of the bar can cause measurement inaccuracies due to the complexity of the geometry. |

Table 6 and Figure 16 show the evolution of the different systems studied in the mass, volume, and moment of inertia. The mass is the same for the four systems, whereas the volume and the moment of inertia of the circular (SHPB-C) and square (SHPB-S) geometry are almost identical, and the hexagon and the triangle have different results.

**Table 6.** Mass of the bars and striker between the different SHPB system.

| Mass (kg) | Circle | Triangle | Square | Hexagone |
|---|---|---|---|---|
| Striker | 1.23 | 1.23 | 1.23 | 1.23 |
| IB | 7.35 | 7.35 | 7.35 | 7.36 |
| OB | 4.9 | 4.9 | 4.9 | 4.9 |

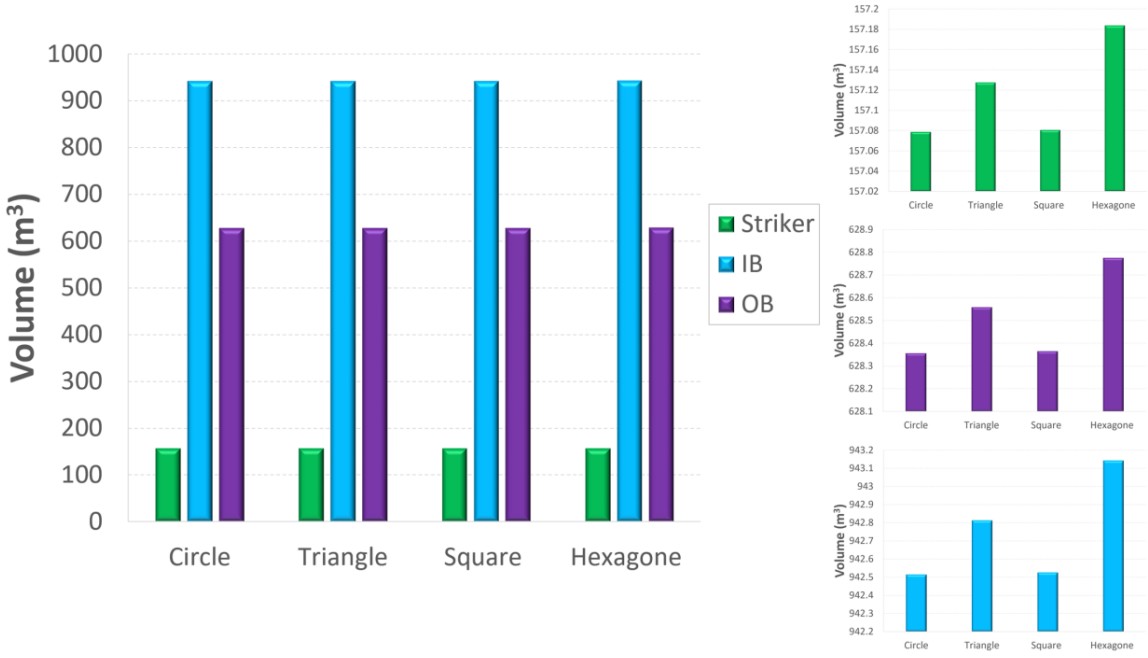

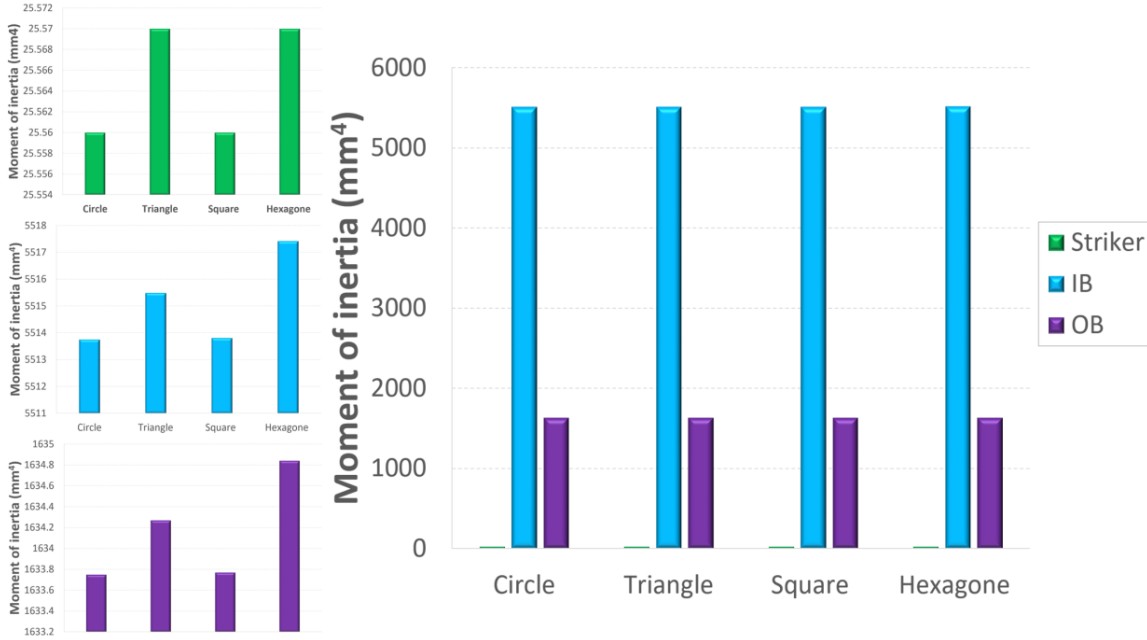

**Figure 16.** Evolution of the geometrical parameters of the SHPB system.

## 5. Conclusions

In this research, we performed a numerical simulation of the high strain rate response of aluminium specimens for different geometric shapes of bar geometries of the SHPB system. Firstly, a numerical model was built in Abaqus software and compared with experimental curves. An excellent correlation between experimental data and numerical results is noted. In addition, the response of the different bar systems is nearly similar. Finally, it may be noted that we have a relatively equivalent distribution for the various SHPB systems.

**Funding:** This research received no external funding.

**Institutional Review Board Statement:** Not applicable.

**Informed Consent Statement:** Not applicable.

**Data Availability Statement:** Data available on request.

**Conflicts of Interest:** The authors declare no conflict of interest.

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
