# Peer review of "Dynamic Composite Materials Characterisation with Hopkinson Bars: Design and Development of New Dynamic Compression Systems"

_jcs, doi:10.3390/jcs7010033_

Round 1

Reviewer 1 Report

The manuscript titled ‘Dynamic composite materials characterisation with Hopkinson 2 bars: Design and development of new dynamic compression 3 systems’ provides some interesting perspectives from authors.

In methodology, authors need to specify such as setting in modeling interface of Abaqus software when authors used finite element analysis.

No information about computational approaches

In title, please correct ‘characterisation’ to ‘characterization’

Author Response

The manuscript titled 'Dynamic composite materials characterisation with Hopkinson bars: Design and development of new dynamic compression systems' provides some interesting perspectives from authors.

Many thanks 

In methodology, authors need to specify such as setting in modeling interface of Abaqus software when authors used finite element analysis.

No information about computational approaches

Changes have been made in the revised version to address these suggestions.

In title, please correct 'characterisation' to 'characterisation'

I am using British English.

Reviewer 2 Report

The manuscript is written with very less grammatical errors but is poorly organized. The introduction section is very confusing. Results and Discussion section contains more information about the mesh procedure. The results presented in this article is not sufficient to consider it for publication. I regret to reject this paper.

The statements in Lines 40, 41 and 42 are a little confusing to read. Please revise them.

Can the authors explain briefly about the new challenges which are mentioned in Lines 49 to 52?

The second paragraph of the Introduction does not seem to have a good flow of information. Several information is compressed in one sentence, and it does not connect with the subsequent sentences. The information provided are not self-explanatory. I suggest the authors to rewrite the whole paragraph and give brief information about each of the information they provide about materials, their strain rate responses, and so on.

In Line 67, please provide a brief information about the guidelines presented by David and Hunter [25].

In Line 70, which configuration is being discussed? Square cross-sections, or circular, or both?

In Line 78, diameter-to-thickness ratio is indicated by (t/d), whereas in previous instances the diameter is indicated by D in L/D ratio. Please keep the symbols consistent throughout the article. If they refer to two different diameters such as Outer and Inner, please explain them. Please explain clearly if this t/d ratio refers to the pulser shape or the SHPB cross-section.

In Line 90, does the form refers to the cross-section of the SHPB?

What are SHPB signals? It must be clearly reported in Line 109.

Please include the symbol for nominal strain rate in Line 114.

There is no information about the numerical modelling of the experiment. Please include a subsection under Section 3 dedicated to explaining the numerical modelling procedure and include some interactive images for explaining the FE modelling. Right now, the discussion seems to jump from materials to FE results without any explanation. Including this section will give more information about the absence of damper bar in the global model, etc.

Paragraph starting from Line 144 is confusing. Why is the dynamic characterization of aluminium sample presented and discussed here? It is mentioned at the beginning of the section that the material used is s 2400 Tex E-glass fibres impregnated with an epoxy polymer resin. Please explain this clearly.

Figure 3 is not self-explanatory. Several important information is missing. What is Test_Inc and what is Test_Tran? Why FEA_Inc results are shown in the figure (2nd row ,2nd column)? What does that mean? Then why FEA_Tran results are not shown? Why FEA_Inc results are not reported in the [±20] and [90] bars? The results are not discussed at all. More confusingly, the paragraph prior to that discussed something about aluminium sample.

The results presented in Section 4.2 are not sufficient.

Author Response

The manuscript is written with very less grammatical errors but is poorly organised. The introduction section is very confusing. Results and Discussion section contains more information about the mesh procedure. The results presented in this article is not sufficient to consider it for publication. I regret to reject this paper.

The statements in Lines 40, 41 and 42 are a little confusing to read. Please revise them.

OK, done

Can the authors explain briefly about the new challenges which are mentioned in Lines 49 to 52?

The second paragraph of the Introduction does not seem to have a good flow of information. Several information is compressed in one sentence, and it does not connect with the subsequent sentences. The information provided are not self-explanatory. I suggest the authors to rewrite the whole paragraph and give brief information about each of the information they provide about materials, their strain rate responses, and so on.

The second paragraph of the Introduction has been rewritten as suggested by the reviewer. More information is given to overview the work carried out clearly.

In Line 67, please provide a brief information about the guidelines presented by David and Hunter [25].

OK, done

In Line 70, which configuration is being discussed? Square cross-sections, or circular, or both?

This is for a pulse shaper with circular cross-sections

In Line 78, diameter-to-thickness ratio is indicated by (t/d), whereas in previous instances the diameter is indicated by D in L/D ratio. Please keep the symbols consistent throughout the article. If they refer to two different diameters such as Outer and Inner, please explain them. Please explain clearly if this t/d ratio refers to the pulser shape or the SHPB cross-section.

isotropic materials: cylindrical shape with an L/D ratio (L: length, D: diameter of the specimen)

pulse shapers: diameter-to-thickness ratios (t/d) (t: thickness, d: diameter of the pulse shapers)

In Line 90, does the form refers to the cross-section of the SHPB?

Yes, it refers to the cross-section of the SHPB.

What are SHPB signals? It must be clearly reported in Line 109.

A section on the experimental procedure used has been added. Figure 3 gives an example of the incident and transmitted gauge signals.

Please include the symbol for nominal strain rate in Line 114.

OK, done

There is no information about the numerical modelling of the experiment. Please include a subsection under Section 3 dedicated to explaining the numerical modelling procedure and include some interactive images for explaining the FE modelling. Right now, the discussion seems to jump from materials to FE results without any explanation. Including this section will give more information about the absence of damper bar in the global model, etc.

A detailed description of the finite element model has been added.

Paragraph starting from Line 144 is confusing. Why is the dynamic characterisation of aluminium sample presented and discussed here? It is mentioned at the beginning of the section that the material used is s 2400 Tex E-glass fibres impregnated with an epoxy polymer resin. Please explain this clearly.

To show the power of the numerical model developed for the simulation of dynamic tests, I started with a model of an orthotropic (composite) material. The latter requires more experimental data and is relatively more complex to model. Then, to focus on the effect of the bar geometry, I chose an isotropic material (fewer data to provide). Using composites and using the different systems of Hopkinson bars studied (circle, square, triangle and hexagon) can generate stress concentrations and cause the initiation of damage; for example, matrix cracking.

Moreover, the thousands of tests available are essentially focused on the characterisation of composite materials.

Figure 3 is not self-explanatory. Several important information is missing. What is Test_Inc and what is Test_Tran? Why FEA_Inc results are shown in the figure (2nd row, 2nd column)? What does that mean? Then why FEA_Tran results are not shown? Why FEA_Inc results are not reported in the [±20] and [90] bars? The results are not discussed at all. More confusingly, the paragraph prior to that discussed something about aluminium sample.

Details of the EF model have been introduced to provide more clarity.

Concerning "Test_Inc" and "Test_Tran", a nomenclature of Figure 5 has been added.

Here, we are interested in the material's behaviour for low-impact velocity that does not generate damage. FEA_Tran has not been represented because there is an equilibrium with FEA_Inc; therefore, it is the same curve but with a time shift. You can see it in Figure 12 in the revised version. Figure R1 gives the incident and transmitted forces for a circular section.

Figure R1: Incident and transmitted Load, FEA

The results presented in Section 4.2 are not sufficient.

Other results have been implemented in the revised version of this manuscript.

Reviewer 3 Report

In this manuscript, a study has been carried out by developing a finite element model to evaluate the effect of different shapes of the bars on the behaviour of materials and the accuracy of the results. The finite element model of conventional circular cross-sectional bars was first validated with the experimental results and then compared with the other shape bars i.e., triangle, square, and hexagon. The flat surface will not only solve the problem of attaching strain gauges on the surface of bars to ensure more accuracy in results but also result in better real-time imaging of material deformation. The manuscript said that the flat surface will not only solve the problem of attaching strain gauges on the surface of bars to ensure more accuracy in results but also result in better real-time imaging of material deformation. Overall, the behaviour of the 4 bar systems is almost similar. Overall, the behaviour of the 4 bar systems is almost similar. However, the author noticed that for the hexagon section, we have a slight difference with the circular section and this difference amplifies with the triangle section. 

Overall, the angle of this manuscript is quite interesting and provided a new way for doing SHPB. The manuscript is easy to read and the introduction did a decent job of introducing the background. However, how much is the necessity of changing shapes of an existing protocol should be discussed more in the paper to highlight the significance of doing so.

1.         How much is the necessity of changing shapes of an existing protocol should be discussed more in the paper to highlight the significance of doing so. Knowing that using the circular section is a standard way for doing SHPB, not only because having a standard instrument is easier for launching the comparisons between different materials and experiments, but also from the perspective of manufactory angle, circular shape is easier to produce. Thus, more detailed discussion needed in the manuscript, such as specifying some obstacles and more advantages of using different shapes for doing SHPB. Especially from the practical angle, because at the very end, practice in the real world is the ultimate goal of any technique. 

2.         Which kind of properties would be more favourable than other material for choosing another shape of section rather than using the circular section? Specifying the preference might help readers to choose accordingly.

3.         The author mentioned that “we can note that for the hexagon section, we have a slight difference with the circular section and this difference amplifies with the triangle section.” Is this difference a linear change or it obeys some formulable rules with the decrease of the edges of the shape?

4.         Why author choose triangle, square and hexagon but excluded pentagonal shape?it seems reasonable to have a linear increase of the number of edges (from 0 to 3, 4, 5, 6 etc.) and it might reveal some rules of change additionally?

5.         Page 3 line 108, proprieties? properties?

Author Response

In this manuscript, a study has been carried out by developing a finite element model to evaluate the effect of different shapes of the bars on the behaviour of materials and the accuracy of the results. The finite element model of conventional circular cross-sectional bars was first validated with the experimental results and then compared with the other shape bars i.e., triangle, square, and hexagon. The flat surface will not only solve the problem of attaching strain gauges on the surface of bars to ensure more accuracy in results but also result in better real-time imaging of material deformation. The manuscript said that the flat surface will not only solve the problem of attaching strain gauges on the surface of bars to ensure more accuracy in results but also result in better real-time imaging of material deformation. Overall, the behaviour of the 4 bar systems is almost similar. However, the author noticed that for the hexagon section, we have a slight difference with the circular section and this difference amplifies with the triangle section. 

Thank you for these relevant and exciting comments.

Overall, the angle of this manuscript is quite interesting and provided a new way for doing SHPB. The manuscript is easy to read and the introduction did a decent job of introducing the background. However, how much is the necessity of changing shapes of an existing protocol should be discussed more in the paper to highlight the significance of doing so.

In this paper, we trace the various drawbacks of the circular section bars based on thousands of tests performed with this system. So the question we asked ourselves was: what if we changed the geometry of the bars to avoid these problems?

Knowing that the experimental protocol will not be modified. Only the guides should be changed to fit the surface of the bars.

In the revised document, this protocol has been well-detailed.

  1.         How much is the necessity of changing shapes of an existing protocol should be discussed more in the paper to highlight the significance of doing so. Knowing that using the circular section is a standard way for doing SHPB, not only because having a standard instrument is easier for launching the comparisons between different materials and experiments, but also from the perspective of manufactory angle, circular shape is easier to produce. Thus, more detailed discussion needed in the manuscript, such as specifying some obstacles and more advantages of using different shapes for doing SHPB. Especially from the practical angle, because at the very end, practice in the real world is the ultimate goal of any technique. 

The necessity of changing the shapes of an existing protocol for using a circular cross-section of split Hopkinson bars depends on the application. If the application requires a circular cross-section, then it is necessary to modify the protocol. Otherwise, the existing protocol may be sufficient if the application requires a different shape. Circular cross-sections are used for split Hopkinson bars because they provide maximum stiffness and strength while minimizing the amount of material used. The circular shape also allows the bar to be easily split into two halves. The circular cross-section also helps to spread the stress over a larger area, reducing the risk of failure due to a localized point of high stress. Other geometries of a straight section of Hopkinson bars are possible, including curved cross-sections, stepped cross-sections, and hollow cross-sections. However, it is important to note that the design of the bar should be tailored to the specific application, as the geometry of the bar will affect its performance.We tried to bring more clarification to this very interesting question that is why several paragraphs have been added to the manuscript.

Several changes have been made in the corrected version to answer these different questions.

  1.         Which kind of properties would be more favourable than other material for choosing another shape of section rather than using the circular section? Specifying the preference might help readers to choose accordingly.

Which properties would be more favorable than other materials for choosing another shape of the section rather than using the circular section for Hopkinson bars?

Pro

perties such as Young’s modulus, yield strength, and fatigue strength would be more favorable than other materials for choosing another section shape. Materials with a higher Young’s modulus and yield strength are better suited for use in Hopkinson bars because they can withstand the large compressive and tensile forces applied during testing. Materials with higher fatigue strength are also better suited to Hopkinson bars because they can be used for longer periods of time without losing their structural integrity. Additionally, material with a higher elastic modulus will be more favorable for choosing another section shape for Hopkinson bars.

  1.         The author mentioned that “we can note that for the hexagon section, we have a slight difference with the circular section and this difference amplifies with the triangle section.” Is this difference a linear change or it obeys some formulable rules with the decrease of the edges of the shape?

This question could be answered accurately when moving to a classification of straight sections (e.g., using a linear increase in the number of edges (from 0 to 3, 4, 5, 6 etc.) and this could reveal some rules of change in addition.

  1.         Why author choose triangle, square and hexagon but excluded pentagonal shape?it seems reasonable to have a linear increase of the number of edges (from 0 to 3, 4, 5, 6 etc.) and it might reveal some rules of change additionally?

I completely agree with you on this point. However, for this first paper, we have chosen the most used geometrical shapes without forgetting the others .....

Another paper is planned to complete this paper, with the establishment of analytical formulations associated with the change of the cross section of the bars and its impact on the mechanical behavior of the tested specimen. In addition, a section will be dedicated to the consideration of the damage of the specimen which will allow to go further in the analysis of the effect of the geometry of the bars

  1.         Page 3 line 108, proprieties? properties?

OK, done

In summary, paragraphs, tables, and figures have been added.

Thank you once again for reviewing our paper

Reviewer 4 Report

Author presented an interesting paper on the use of different geometries for the Hopkinson bar. The main advantage would be the easer application of strain gages. However, there is no relevant conclusions on that issue. Also, the use of composite specimens was not discussed and no relevant conclusion were obtained.

It would be interesting to have the following information/comments:

- What epoxy system was used and what was the reinforcement form (fabric?, in this case the areal weight should be given)

- what is the meaning of 90º? Is this not equivalent to 0º?

- what was the motivation for choosing the staking sequences referred?

- How did authors assessed the void fraction on composites?

- What was the aluminum alloy used for specimens?

- What were the elastic properties of composite specimens used in the FEM simulations?

Author Response

Author presented an interesting paper on the use of different geometries for the Hopkinson bar. The main advantage would be the easer application of strain gages. However, there is no relevant conclusions on that issue. Also, the use of composite specimens was not discussed and no relevant conclusion were obtained.

It would be interesting to have the following information/comments:

- What epoxy system was used and what was the reinforcement form (fabric?, in this case the areal weight should be given)

A more detailed description of the materials used has been added in the corrected version.

Composite: 2400 Tex E-Glass fibres impregnated with an epoxy matrix.

Resin: EPOLAM pre-polymer, EPOLAM 2020 hardener and 2020 accelerator from Axson.

Reinforcement: fabric with 90% warp yarns and 10% weft yarns.

- what is the meaning of 90º? Is this not equivalent to 0º?

Yes, the 0° and 90° are identical. However, when dynamic tests were carried out, two loading directions were chosen: in-plane and out-of-plane.

For in-plane tests, we have different behaviours for 0° and 90° [1-2].

For out-of-plane tests, 0° and 90° are equivalents [3].

  • Tarfaoui, M., Nême, A., & Choukri, S. (2009). Damage kinetics of glass/epoxy composite materials under dynamic compression. Journal of Composite Materials43(10), 1137-1154.
  • Tarfaoui, M., Choukri, S., & Nême, A. (2010). Dynamic response of symmetric and asymmetric e-glass/epoxy laminates at high strain rates. In Key Engineering Materials(Vol. 446, pp. 73-82). Trans Tech Publications Ltd.
  • Tarfaoui, M., Choukri, S., & Nême, A. (2008). Effect of fibre orientation on mechanical properties of the laminated polymer composites subjected to out-of-plane high strain rate compressive loadings. Composites Science and Technology68(2), 477-485.

- what was the motivation for choosing the staking sequences referred?

The choice of composite depends on the field of application. Under this project, the study focused on marine structures (the behaviour of tubular structures (AUVs) and frigate superstructures).

- How did authors assessed the void fraction on composites?

We based ourselves on the standard, which is based on microscopic analysis and the development of a Matlab program which allows the processing of images.

- What was the aluminum alloy used for specimens?

It is aluminium 6014-T4. A note has been added in table 2.

- What were the elastic properties of composite specimens used in the FEM simulations?

Table 2 also shows the elastic properties of the composite specimens used in the FEM simulations. 

Round 2

Reviewer 2 Report

The authors have made a significant improvement to the article. It can be now accepted in its present form. 

Reviewer 3 Report

The new version of manuscript seems improved from the first version with the related topic.

Reviewer 4 Report

The paper allows to conclude that different shapes for the Hopkinson bar can be used, namelly with some composite materials. The revised paper has made conclusions more clear.